# Fruit Biology of Coconut (*Cocos nucifera* L.)

**DOI:** 10.3390/plants11233293

**Published:** 2022-11-29

**Authors:** Fernanda Caro Beveridge, Sundaravelpandian Kalaipandian, Chongxi Yang, Steve W. Adkins

**Affiliations:** School of Agriculture and Food Sciences, The University of Queensland, Gatton, QLD 4343, Australia

**Keywords:** coconut, germination, palms, propagation, seed

## Abstract

Coconut (*Cocos nucifera* L.) is an important perennial crop adapted to a wide range of habitats. Although global coconut demand has increased sharply over the past few years, its production has been decreasing due to palm senility, as well as abiotic and biotic stresses. In fact, replanting efforts are impeded due to the lack of good quality seedlings. In vitro technologies have a great potential; however, their applications may take time to reach a commercial level. Therefore, traditional seed propagation is still critical to help meet the rising demand and its practice needs to be improved. To achieve an improved propagation via seeds, it is important to understand coconut fruit biology and its related issues. This review aims to provide a comprehensive summary of the existing knowledge on coconut fruit morpho-anatomy, germination biology, seed dispersal, distribution, fruit longevity and storage. This will help to identify gaps where future research efforts should be directed to improve traditional seed propagation.

## 1. Introduction

Coconut (*Cocos nucifera* L.) is a monoecious perennial palm from the Arecaceae [1]. It is native to the present-day humid tropics (between latitudes 26° N and 26° S of the equator [2]) and adapted to a wide range of habitats, from shorelines to the foothills of mountain ranges in more than 200 countries [3] and it is grown as a crop in a total of ca. 12 million ha of cultivated land around the world in over 90 countries [4,5]. The centre of its origin and the extent to which it naturally disperses is still under debate [6]. It has been hypothesised that the coconut palm evolved in coral atoll ecosystems, from where its fruit dispersed by floating on the sea, to new islands; a process which imposes a high selection pressure on specialist survival and germination process of the newly arrived coconut fruit [6]. More than 80% of the coconut production comes from the Asia-Pacific region [7]. This places coconut as one of the most important sources of income for many countries in this region, having valuable export, local economic and cultural impact [4].

Coconut is called as the “tree of life” since every part of the plant is useful [4]. With a great versatility, coconut provides a bountiful source of food, household products and income for ca. 11 million small holder farmers around the world [8]. In most countries, coconut farming has been focused on copra production to obtain crude endosperm oil and dried kernel. However, more recently there has been a shift to produce more lucrative products, including those with high nutritional value and medicinal uses [9], including shell charcoal, husk fibre, virgin oil, as well as coconut wood for production of furniture, building materials, veneer and handicrafts [10]. This range of diverse products places the coconut palm as one of the most important crops in tropical countries with an annual total production of ca. 61,165 million nuts or 13.59 million tonnes of copra equivalent [8].

Coconut farming is currently facing many challenges including palm senility, devastating abiotic and biotic stresses, market instability of its products and the insufficient production of seedlings to replant the ageing palms [10,11]. In addition, coconut producers must deal with the decline in annual fruit production as about 50–70% of the presently cultivated coconut palms are well past their production peak age and are regarded as senile and unproductive [11]. There is a steady decline in fruit production after a coconut palm reaches 35 years of age, due to a reduced ability to extract soil nutrients, a decrease in leaf area and accumulated damage due to various abiotic factors such as droughts, storms, cyclones, and tsunamis [5]. Moreover, several major pests and diseases (e.g., the coconut rhinoceros beetle [*Oryctes rhinoceros* L.], the cadang-cadang viroid, and the lethal yellowing phytoplasmas) have posed a significant economic impact on the coconut industry [4,12,13]. These challenges, coupled with the fact that most coconut farmers have limited access to high-quality planting materials and have a low desire to adopt new technologies, have resulted in a significant decline in total growing area [14] and pose serious threats to the future of the coconut industry.

Difficulties in conventional methods for seedling production start with the palm’s long juvenile phase [15], the low planting density [16], the low and slow rate of fruit production that requires hand pollination to take place, and the variability in seed germination and lack of knowledge on its triggers [17]. Due to these issues, in vitro propagation approaches designed to produce cloned plantlets, rapidly and in vast numbers are being developed as alternative approach to produce planting materials. Even if tissue culture technologies are commercialized, fruit-based seedling production is still necessary to meet the ever-increasing need for seedlings. Hence it is necessary to improve the seed propagation techniques in coconut to increase the production of seedlings.

Although attempts have been made in the past to improve coconut fruit germination for conventional crop propagation [18,19,20,21,22], little attention has been given to the fruit-based propagation techniques in recent times. In addition, there is a lack of contemporary published studies on the biological characteristics of the coconut fruit, the production of fruits and methods to improve and hasten its germination. The coconut crop has not been domesticated to the same extent as many other crops, still retaining many features from its wild ancestors or its partially domesticated predecessors [3]. Many present-day palm varieties grown possess characteristics like those of wild palms, such as low and slow seed set, and non-uniform seed germination. The coconut fruit is a big, oily [23] fruit containing a single recalcitrant seed [24], unable to tolerate desiccation and therefore having a limited storage life. Due to the large size of the fruit, it is difficult to perform glasshouse experiments on germination [16] and field experimentation requires a vast number of resources [25]. Hence, few attempts have been made to study the coconut fruit germination.

To date, sexual reproduction from seeds is the only propagation method available for coconut [26]. However, the germination requirements, the methodology to improve germination and their dormancy characteristics have been poorly studied. Hence, this paper attempts to review the available literature on the fruit dispersal and seed germination biology, and to identify gaps that need to be studied for the benefit of the current coconut industry which still relies on replanting of senile and diseased palms through germinating fruits. This review provides a comprehensive summary of the existing knowledge on coconut fruit by describing (1) the fruit morpho-anatomy, (2) fruit distribution and dispersal, (3) seed germination and dormancy including defining a consistent dormancy classification for this species and describing germination methods used in fruit germination, (4) fruit longevity and storage and (5) identifying gaps in the field where future research efforts should be directed to improve coconut propagation through fruit germination.

## 2. Morpho-Anatomical Characterization of the Coconut Fruit

### 2.1. Morpho-Anatomy

Coconut palms produce a large, single seeded fruit [21] (Figure 1), referred to as a drupe [27]. The fruit consists of a thin outermost layer (the exocarp or epidermis, 0.1 mm thick), a thick layer of fibrous tissues referred to as the husk or coir (the mesocarp, 1 to 5 cm thick) and a hard lignified layer or shell (the endocarp, 3 to 6 mm thick) [1,21,28]. The hard endocarp provides protection from predators and the fibrous mesocarp protects the nuts when falling from trees [29] and aids flotation. A thin brown seed coat layer (the testa) adheres beneath the endocarp [1]. The solid endosperm within the testa varies in thickness from 0.8 to 2.0 cm [21], weighing from 98 to 553 g fresh weight. Within this is the water or liquid endosperm [21,27] which ranges in fresh weight from 21 to 449 g [30] depending on the variety, management practices and environmental factors at the time of fruit development.

The coconut endocarp carries three longitudinal ridges on its surface, which are a result of the fusion of the three carpels. Three eyes or pores at the peduncle end of the fruit and between the ridges are present [31]. One of the eyes is soft with weakly lignified tissue, under which the small and cylindrical embryo is embedded within the solid endosperm [23]. The embryonic shoot will emerge through the soft eye, which is known as the ‘germ pore’ [21]. The endocarp wall at the point of the soft eye has branched vascular tubes, which conduct sap through the peduncle, feeding the developing embryo and endosperm during fruit development [26]. During the germination process, these vascular tubes can also allow moisture to penetrate to the embryo [32]. The solid endosperm forms after 210 days following fertilization, initially from opposite to the stem end [33], then it progressively extends around the internal cavity to surround the embryo and achieves its maximum thickness at 360 days after fertilization when the fruit becomes physiologically mature. The liquid endosperm or coconut water usually reduces its volume after maturation [26], and the reason is still unknown. When the water reduces, the space might be replaced with gases or vacuum, and the splashing sound cannot be heard when a coconut is shaken (a common practice used to determine coconut maturity).

### 2.2. Outer Fruit Morphology

The coconut fruit is diverse in its outer morphology including its shape, colour and size [34] (Figure 2). For example, fruits are classified as either being elliptic, oblong or round by shape, and yellow, green, orange, brown or red by colour [1]. The immature fruit from tall varieties usually displays different intensities of green and bronze colours, with yellow colours occasionally being found in the Pacific region. On the other hand, the fruit of the self-pollinating dwarf varieties can display orange, yellow, green and light brown colours [9]. Further, the biomass proportions of husk, shell and kernel biomass can show large genetic diversity in the species [35]. The wild population are believed to have naturally evolved with an increased proportion of husk and a long and angular shape [18], as such traits favour the dispersal of fruits by water. In addition, wild coconuts are usually found to germinate slowly, and have about 25% of their biomass in the endosperm, 15% in the shell and 60% in the husk. These characteristics promoted buoyancy, while the thick husk ensured the embryo was protected from sea water [18].

It is believed that coconut was first domesticated in the larger islands and along the continental coast of Malesia [36], where coastal fishing communities used naturally dispersed coconuts as a source of water [37]. Domestication favoured a spherical shaped fruit (Figure 3) with a concomitantly reduced husk thickness (from 70 to 35% of total nut fresh weight), and an increased water content and germination rate [18]. Round shapes were desired as they can hold larger volume of water in the cavity and are easier to stack in boats [27]. Domestication might have also favoured higher water content, which resulted in less mesocarp content and a higher fruit density, and decreased buoyancy [38]. Such trait selections caused the alteration of biomass of about 50% in the endosperm, 20% in the shell and 30% in the husk [18]. As a round shape was the desired fruit trait, width would have increased considerably which in turn increased the kernel size by 25 to 30% [39]. Domestication created differences between wild and domesticated types including plant habit, disease resistance and wind-storm tolerance [18,19,36,40] as well as fruit size, shape, and yield among modern coconut varieties [6]. Introgression of favourable traits from wild trees to domesticated varieties has led to intermediate biomass proportions of the fruit components in coconut [11]. The only component which has remained stable irrespective of domestication and environmental conditions is the endocarp, which is about 12 to 16% of the fruit [18]. Fruit morphology might also be influenced by the application of fertilizers. Studies undertaken in Indonesia and the Philippines suggest that applying potassium or sodium chloride can result in a higher solid endosperm content (Magat, pers. comm.).

### 2.3. Embryo Anatomy

The coconut fruit has a small embryo-to-seed ratio (E:S) and a large quantity of endosperm tissue like most palm fruits [41,42]. The endosperm is usually about 100 times the weight of the embryo in mature coconut seeds [43]. Morpho-anatomical experiments during embryo development have been poorly studied in many palm species including coconut. When mature, the coconut embryo has a cylindrical shape of ca. 0.8 cm long [1]. Palm embryos are histologically composed of a short plumular-radicular axis (epicotyl with leaf primordia and hypocotyl-radicle axis) and a large, single cotyledon at the proximal end of the cotyledonary tube [1,44]. In coconut embryos, the simple disc of embryonic cells will differentiate into a single cotyledon, and then it differentiates into a tubular base, consisting of a petiole and a distal haustorium [45]. The plumule consists of a central meristematic zone enclosed by scaly-leaf primordia, which are surrounded by the cotyledonary petiole. The radicle is present opposite to the plumule within the apical mass of meristematic cells positioned near the suspensory region, and it usually differentiates slowly, and can be identified when the primordium grows [46]. Most palm species have similar embryo structure and anatomy, and the major difference is the disposition of the embryo axis which can be oblique or parallel to the cotyledon [47], being oblique in coconut [46]. A haustorium forms and occupies the space within the endosperm cavity and both the liquid and solid endosperm progressively disappear from the cavity [41].

From the above studies, it is evident that the morphological and anatomical characteristics of the coconut fruit are poorly understood, and further studies could help to uncover its seed germination process, which will in turn be beneficial to increase the production of high-quality coconut seedlings from fruits for re-planting of coconut palms around the world. Coconut often takes from 60 to 220 days to achieve 90% germination [27]. The typical production of healthy coconut seedlings from fruits is highly variable depending on variety and methods of germination and growing conditions. Considering the poor germination and time taken to achieve higher percentage of germination of several market-demanded coconut varieties, it is important to understand the germination mechanism of coconut as it will substantially contribute to global demand for good quality coconut seedlings.

## 3. Coconut Distribution and Dispersal

### 3.1. Sea Water as a Dispersal Agent

The centre of origin for coconut is not well-established but it is believed that it may have originated in the ancient supercontinent of Gondwana [48]. The Americas and Asia have been proposed as the geographical origins, but these possible geographical locations have not been ascertained [6]. Although the centre of origin is unknown, the coconut palm continued to evolve to produce highly successful buoyant fruits and dispersed between land masses by water [3]. Thus, coconut can disseminate by itself under natural conditions, mainly by floating in sea water [6,28]. As the coconut fruit is large and heavy, it is not suitable for dispersal by animals [27]. Therefore, ocean drift could be the most important factor for natural dispersal of coconut in the present-day Indo-Pacific regions [49]. The wild coconut fruit is capable of surviving for ca. 110 days in the ocean and this was estimated to float 4800 km [50]. In this way, coconut would have dispersed several millions of years, prior to human colonization [51]. Coconut colonized numerous scattered islands, including coral atolls and recently formed volcanic islands, by floating [6], where it could establish without competition from other plant species and herbivorous attack [52]. Harries and Clement [6] have proposed that coconut fruit floating duration is more important than floating distance as the viable fruits fallen into the lagoon would often replace the palms that were destroyed by natural calamities or establish on new coral islands, and the long-distance floating might have been a less frequent event.

At this point in time, without human assistance, coconuts would be limited to grow only on the narrow shorelines around the Indo-Pacific basin [27,49]. Once on a new coral atoll, palms could establish with further limited movement along the shoreline [38]. Establishment of coconut palms on shorelines is favourable as a freshwater lens is trapped above the saline groundwater [19] and prevents exposure of the palm to extended dry periods [38]. At much later times, this species has been more widely dispersed by humans to many other suitable habitats within tropical regions around the world, including the coasts lapped by the Atlantic Ocean [51]. The exact period of redistribution of coconut populations through human action is unknown, but it is believed that it could be the time (>100,000 years) when human colonisation of tropical coasts was first being undertaken [53].

### 3.2. Fruit Adaptations to Sea Water

The coconut fruit and its embryo have evolved with interesting adaptive traits to survive in long-distance dispersal by oceanic currents, to later germinate on sandy beaches [6,41,50]. Possibly, the wild fruit prior to domestication had a thick fibrous mesocarp with a hard stony endocarp that protected and preserved the embryo viability while floating on salty water [6,27]. Thus, the embryo development within the fruit would have been largely isolated from harsh outside environmental conditions over several months, which favoured the long-distance dissemination of the fruit by ocean currents. Although the mesocarp was thick, the wild coconut fruit had only a small, 12 mm thick solid endosperm but had a large buoyancy cavity partially filled with water to allow it to float, unless the husk got saturated with water [6,41,50].

For successful dispersal in sea water, embryo tolerance to the de-hydrating effects of salt water is required, and the salinity tolerance of the embryo varies amongst varieties [16]. Despite the embryo’s tolerance to salinity, if the fruit germinates while floating, the emerging seedling cannot tolerate sea water and will not survive for long [18]. Hence, the delay in germination time will consequently limit the travel distance of the fruit. Therefore, it has been suggested that slow rates of germination would enable successful inter-island or trans-oceanic dispersal [18]. After the fruit arrived on a beach, the angular fruit shape (which also allowed for wind dispersal on the ocean) would result in the fruit being moved to the top of the beach and then prevent it from rolling away back into the ocean. This allowed for rooting to take place at the top of the beach, having a greater chance to access fresh water for successful germination [27]. Harries [27] proposed that sea water absorbed through the husk would increase the matric potential around the embryo, which would induce further mechanisms for slowing the germination process, however, this hypothesis has not been tested. If this hypothesis is true then it is likely that once the fruit is in contact with fresh water, this inhibiting mechanism would be reversed, and germination would be promoted. In summary, understanding the selection pressures encountered by the coconut fruit in its natural environment and the sequence of events from the time of fruit fall in one habitat to germination in a new habitat, would provide the mechanisms that controls germination of coconut fruit on both a physiological and ecological level.

## 4. Seed Germination and Dormancy

### 4.1. Germination Process

The germination process of coconut seeds, as in most palm species, occurs in an unusual manner [41,54,55] and does not follow the standard definition of germination involving early radicle protrusion [56]. The fruit takes between 330 to 420 days to develop after pollination, before germination can occur [57]. During germination (Figure 4 and Figure 5), embryo growth occurs simultaneously in two directions. In the proximal end of the embryo, the cotyledonary petiole grows through the germ pore, containing the plumule and radicle, which then grow outside of the endocarp. The cotyledonary blade, located on the distal end of the embryo, expands to form a sponge-like, pear-shaped specialized haustorium which fills most of the endosperm cavity [1,41] in a period of 140 to 180 days after germination starts [58]. The haustorium only starts expanding after the cotyledonary petiole has emerged from the seed. This sequence of events means the embryo needs to undergo a development period outside the seed prior to the completion of the germination process (if germination is to be defined by radicle emergence, *viz*. [56]). The requirement for embryo tissue development post germination places palm embryos (including coconut) as having a special type of underdevelopment [42]. The standard definition of an underdeveloped embryo is when the embryo needs to grow inside the seed before radicle protrusion (germination) can take place [42].

Three types of germination patterns have been observed in palm species, *viz*. remote tubular, remote ligular and adjacent ligular. Coconut embryos have an adjacent ligular-type of germination, which involves the formation of a structure called the germinative button [28,47,59]. Once the plumule emerges from the germ pore, it can start to photosynthesize and promotes the development of a strong rooting system, but this stage is still maintained within the mesocarp and protected from outside environmental conditions [21]. The endosperm remains the only source of energy for growth of germinated embryos prior to shoot emergence and no significant development of leaf area occurs for ca. 60 days, as the shoot takes this time to pass through the thick husk [21], and the time depends on the husk thickness [27]. After the emergence of the shoot, both the endosperm reserves and photosynthesis by the shoot will supply energy for the next 360 days of growth [52]. Although coconut germination has been defined above from a biological perspective, true germination can only be observed in de-husked fruits [22,60]. Therefore, coconut growers have developed their own agronomic definition for germination that is the emergence of the plumule from the husk [18,22,60].

The haustorium is initially involved in the regulation of the uptake of nutrients from the endosperm [41,61]. However, the principal role of the haustorium is in the absorption of oils (such as triacylglycerol and free fatty acids) [58], released during endosperm degradation, and then it converts oils into sugar storage reserves to supply energy to the growing seedling [23]. It has been reported that the liquid endosperm (coconut water) is broken down first to supply the energy and then the solid endosperm breaks down [62,63]. Mazzottini-dos-Santos, et al. [64] studied the role of the haustorium in the macaw palm (*Acrocomia aculeata* (Jacq.) Lodd. Ex Mart.) and found that there was no cytological evidence that this structure secretes enzymes that can mobilize the breakdown of the endosperm. Thus, the structure only functioned as an absorption organ for the temporary storage of energy-rich substrates. López-Villalobos, Dodds and Hornung [61] compared embryonic tissue growth and fatty acid composition in in situ-germinated coconut seedlings and in vitro-germinated seedlings from zygotic embryos. In situ seedling tissues had medium and long chain fatty acids as well as a well-developed haustorium, but in vitro-germinated seedlings did not develop a significant haustoria and failed to accumulate lauric acid without access to endosperm tissues. They found a correlation between the accumulation of lauric acid and the development of the haustorium in the in situ seedlings and suggested that the haustorium had a key role as a site of operation of the glyoxylate cycle, which helps convert lipids into sugars.

There is a wide variation in the germination time among cultivated varieties and wild coconuts ranging from 30 to 220 days [17], which has been assumed to be under both genetic and environmental control [17,28]. There are two types of germination pattern found in coconut populations around the world. The first form is the late germinators that take >75 days to achieve 25% germination, >90 days for 50% and >105 days for 75%. The second form is the early germinators, and those kinds require less time to reach the same percentages [19]. There are two types of coconut palms based on their height, tall and dwarf types. In tall types, female and male flowers open at different times which leads to cross pollination, on the other hand in dwarf types, both flower types open at the same time, so self-pollination occurs. Tall varieties will usually germinate from 60 to 200 days after sowing, with maximum germination percentages being reached between the 119th and 126th days [60]. Dwarfs will germinate between 30 to 95 days after sowing, and rarely, viviparous germination can happen [1]. Once the fruit starts the germination, seedling establishment and survival will be determined by the amount of endosperm present in the seed [18], and the speed at which the seedling can start producing energy via photosynthesis.

These significant differences in coconut germination time are thought to have evolved as a consequence of selection pressures coming from both natural and domestic sources, and such diverse selection could be the reasons why there is a wide range of diversity among coconut varieties [27]. Early and late germinating coconuts have been found to have distinct patterns of global distribution [39], and these distribution patterns have been used as one of the important characterizations of coconut varieties. Such germinating patterns are a useful tool to study the presence of variability between the coconut populations [19]. Late germinating varieties have thicker husk, which might be one of the useful characteristics while floating [18], and domestication favoured thinner husked fruits with increased volumes of coconut water [18]. Therefore, thinner husks are usually related to more rapid germination. Harries [27] proposed a correlation between the early germination (30 to 140 days) trait and the thinner-husk traits of fruits. A spherical-shaped, thin-husked fruit from tall and dwarf varieties have been shown to germinate within 30 days [18]. Further, early germinating fruits have also been associated with early opening of the germ pore, together with a rapid shoot elongation rate [22]. It is possible that the thick husks may impose a mechanical constraint to embryo growth, and this may delay the cotyledonary petiole from protruding through the germ pore. Mechanical constraint imposed by the endosperm and/or pericarp tissues is a common mechanism found in seeds that helps to delay their germination, particularly in seeds with a small embryo-to-seed ratios. On the other hand, Foale, et al. [65] proposed that the husk plays no role in the germination process, and it is confined to protecting the fragile emerging shoot. As true germination can only be seen in de-husked fruits, it is hard to compare germination rates in husked and de-husked fruits, and this issue has not yet been addressed.

Studies have shown that the early germination traits might be positively correlated with an improved chance of survival when disease might be a problem among the population. For example, studies conducted in Jamaica revealed that early germination could be linked to lethal yellowing disease resistance [66]. From the studies of various coconut populations, Zizumbo-Villarreal and Arellano-Morín [38] also proposed that there is a correlation between the rapidly germinated seedlings and disease resistance, and the early germinated seedlings lead to healthier and superior palms. Further research should be focused on the influence of the fruit husk on germination, and the correlation between early germination, seed vigour and seedling survival.

### 4.2. Germination Cues in Nature

Although differences in coconut germination patterns are well known, the factors that influences the germination process are still poorly understood [17]. The ideal temperature for growth and fruit yield of coconut palms is between 27 to 32 °C with a diurnal variation equal to or less than 7 °C in humid environments [2,67]. Coconut fruits can germinate even when positioned in any orientation. After falling from the palm to the ground naturally, fruit settle with one of their broader surfaces touching the ground [1], a position from which they naturally germinate. Similarly, horizontally sown fruit in nurseries have also been shown to give higher germination rates in comparison to vertical sowing [2]. The first cue to trigger germination is the uptake of fresh water into the germ pore. Water needs to enter the mature, dry husk and reach the embryo before germination starts. The germinating fruit has sufficient nutrients, carbon sources and growth regulators for the initiation of germination and early seedling growth and development [1]. In addition to all these traits, fresh water could be the only missing factor that triggers germination under natural conditions.

It has also been proposed that once the fruit reaches the shoreline, the heat of the sand promotes embryo development and subsequent germination [28]. If this were to occur, the temperature of the coconut water in the internal cavity would also increase. To date, no study has been undertaken to understand the effects of heating on the liquid endosperm and on embryo viability and seed germination rates. Furthermore, the husk moisture content, water content of nuts and distance between water and haustorium have been suggested as the reasons for late germination [68]. If the liquid endosperm is exhausted before the full expansion of haustorium, then the seed cannot germinate [69]. From the above studies, it is inferred that the coconut germination process is poorly understood, and future research needs to focus on the genetic and environmental factors affecting coconut germination, such as the maternal effects that can induce different degrees of fruit maturation, the fruit maturity at the time of harvest, and the environmental conditions from harvest to storage [21]. Such studies should focus on both fruit anatomy and biochemistry to gain a mechanistic understanding on the coconut germination process.

### 4.3. Fruit Dormancy

There are inconsistencies in the literature regarding the presence of dormancy in the coconut fruit, and various criteria have been used to define coconut dormancy. Several studies state that coconut fruit do not have dormancy and that they will readily germinate under warm and humid conditions [18,27,49,52]. Nonetheless, even the early- or fast-germinating varieties usually take at least 30 days to germinate. According to Baskin and Baskin [56], viable seeds are classified as being dormant if, when physiologically mature and freshly collected, seeds take more than 28 days to germinate when tested under a range of different environmental conditions. Moreover, several palm species have been described as having an underdeveloped embryo, which needs more time to undergo development after shedding from the mother palm before germination, and to complete the germination process [42]. Since the presence of the above-said traits, it can be assumed that dormancy is present in the coconut fruit as it requires a long time (>28 days) to achieve germination of viable, mature, and freshly collected fruit.

It has been proposed by Baskin and Baskin [70] that coconut fruit might have a morphophysiological dormancy (MPD), as do most palm seeds from the Arecaceae family [71]. Morphophysiological dormancy occurs when seeds take longer than 30 days to germinate and the seed has an underdeveloped embryo. Palm species with late germination have been suggested to have a morphological dormancy (MD) associated with embryo anatomy (underdeveloped embryo), and/or physical dormancy (PY), present in the structures surrounding the embryo [42]. Studies undertaken on the endocarp and seed coat anatomy of different palm species have shown anatomical differences when compared to non-palm species which have PY [42,71]. Tissues covering the embryo in many palm species are known to impose a mechanical obstacle to germination rather than being an impermeable layer to water and oxygen uptake [28]. Consequently, palm seeds do not present PY or combinational dormancy, which is physiological dormancy (PD) combined with PY [42]. Foale [22] proposed that populations of the Niu Kafa (Samoan wild type) displayed an intrinsic period of dormancy after maturity [22]. This could be related with having to float long periods of time on the sea before reaching a destination where the fruit can germinate and successfully establish [22]. The presence of dormancy in coconut is inferred to be an ancestral trait which might have been reduced during domestication.

Pérez, et al. [72] proposed that PD should be overcome prior to overcoming MD in palm seeds. Warm and moist conditions would help overcome PD, increasing the growth potential of the embryo which would dislodge the operculum and rupture the endocarp. Once freed from the endocarp, the embryo can continue to develop and produce a radicle [71]. Therefore, dormancy alleviation could be undertaken in palms species by burying mature and intact seeds in the soil under moist, warm conditions [72]. Warm stratification (25 to 35 °C) for ca. 30 days, removal of part or all the mesocarp, and operculum removal have been proposed as strategies to promote germination rates and final germination in palm seeds [71]. Storage in the shade for 30 days (seasoning) was described as essential to overcome dormancy in coconut fruits [73], and this time could be used for embryo development (therefore overcoming MD). Similarly, [74] obtained higher quality seedlings when fruit were stored 30 days in open shade, then sand cured for 60 to 90 days (with storage time shorter for dwarfs).

In summary, it is important to investigate if MPD could be a common trait, or if it is specific to certain varieties, such as wild types. Identifying the dormancy class present in coconut fruit could improve seedling production from fruit, eliminate unnecessary pre-sowing treatments, and help to determine the optimum storage period and conditions for coconuts. To classify dormancy appropriately, experiments should be undertaken in mature and freshly collected coconut fruits, where embryo growth should be measured from collection until germination to examine if embryo growth occurs throughout this period. Furthermore, examining how water moves inside the fruit and the resistance by fruit and seed tissues to this movement (if any) would help to identify the unlikely presence of PY (based on the literature on other palm species), and provide a clear understanding of the germination process of coconut. If the dormancy class of coconut is known, it can be used to develop germination-enhancing methods to accelerate the germination process, as well as find ways to maintain this dormancy to prolong fruit longevity.

### 4.4. Alleviating Dormancy and Promoting Germination

Although coconut has been used in subsistence cultivation for some millennia and grown in plantations for over a century, germination rates and nursery practices differ between the Pacific and Indian Ocean communities, and within these regions [27]. Hastening and improving total fruit germination and seedling emergence of coconut varieties could have practical value, particularly in areas where tissue culture will not be accessible soon. Besides reducing production costs, reducing the germination time of a seed batch in nurseries would reduce the risk that temporal variability can cause misperceived observation of intrinsic seedling growth performance [22].

Some pre-treatments have been used to hasten and improve the final germination percentage of coconut fruit, including de-husking, then soaking in water or in nutrient solutions (Table 1). Also, chopping the husk from both sides [60], slicing a portion of the husk away [2,65] or completely de-husking the fruit [27] have been proposed. Trimming at least some proportion of the husk above the germ pore has been suggested to stimulate germination by improving moisture penetration through the germ pore [2]. Foale [22] exposed de-husked fruit to 0, 14 and 28 days in an unsaturated atmosphere or low relative humidity, then transferred them to a saturated atmosphere in sacks for germination. Pre-exposed fruit had a more rapid earlier germination than untreated nuts, with nuts exposed for 28 days having the highest germination rate, then followed by the 14-days treatment, then the control. However, all treatments reached the same final germination after 70 days. Foale [22] suggested that the speed of germination was greatly enhanced after exposure. The author proposed that germination was inhibited by exposing the germ pore to a vapour-unsaturated atmosphere, but once those treatments were placed in a saturated environment, they readily germinated. Moreover, de-husking the fruit did not cause damage to the developing seedlings when correctly applied [22].

Soaking the fruit in water for up to 15 days can increase the rate of germination, but if this period is extended both the germination rate and the seedling quality can be impaired [2]. The promoting effects of soaking could be related to ensuring a fast water uptake by the fruit as water is reaching the embryo earlier than if the fruit is unsoaked. It is also possible that soaking acts as a warm stratification treatment, therefore helping to overcome PD. In an experiment conducted in Tanzania, coconut fruit soaked in water had a fastest germination (ca. 81 days) compared with untreated fruit (ca. 143 days) [75]. Germination was also hastened by soaking fruit in solutions of 0.01 M potassium nitrate and 0.02 M sodium carbonate for 48 h [75]. Potassium nitrate is known to act as a germination stimulant, stimulating seeds to operate the alternative pathway of respiration, providing for a better environment for tissue growth [76], or providing the embryo with nitrate (NO_3_^−^) [77]. Potassium nitrate may also act as a signalling molecule [78]. Similarly, Harries [27] has also proposed that de-husked fruits may be soaked in water or in a nutrient solution to improve their germination rate. Other suggested treatments include injecting different minerals and nutrients such as zinc and copper into the fruit [2]. Moreover, the use of gibberellic acid (GA_3_) may be able to promote palm seed germination by weakening the micropylar endosperm tissue surrounding the proximal side of the cotyledonary petiole, and by increasing the embryo growth potential [42]. This plant growth regulator is considerably expensive to use in the coconut nursery and therefore is not recommended as a cost-effective approach to promote germination.

Warm stratification is commonly used to increase the speed of germination of palm seeds [79], with high temperatures required to overcome dormancy, but not essential to promote germination [28]. Warm stratification, as a treatment to overcome PD, could be applied to coconut fruits, particularly to the slow germinating varieties. Temperatures between 27 to 32 °C (optimum temperatures for coconut growth [2]) could be used as a guide to possible stratification treatments. Identifying practices that could accelerate the germination process in coconut fruits to increase the fruit germination could have important economic impacts. To propagate coconut commercially, fruits are first germinated in a nursery, and then (after 300 to 360 days) seedlings are transplanted to the field. Early, more uniform, and higher germination rates would shorten the nursery period and reduce fruit waste, thereby reducing the cost of production [75].

## 5. Fruit Longevity and Storage

It is not always possible to plant coconut fruit right after harvesting. Therefore, it is important to understand the period of storage that fruits can tolerate without having their viability affected. Together with the optimum storage conditions to prolong longevity, it is important to reduce fruit germination during storage. Coconut is considered a recalcitrant seeded species [24,27], a common trait of species coming from tropical, humid environments. Recalcitrant seeds cannot be stored for long periods of time as they cannot tolerate drying or freezing, and this limits their storage time. Prolonged periods of seed storage severely affect germination in some coconut varieties [2]. However, in some cases when farmers must wait for monsoon rains, storage may have to be up to 120 days, which can compromise viability [2]. It has been reported that some coconut varieties can only be stored for 30 days in the shade without losing their viability [74]. Some slow germinating varieties, such as Polynesian and West African Tall, can be successfully stored for 30 days if the liquid endosperm is not allowed to dry out [80].

Recalcitrant seeds remain metabolically active and with high seed moisture contents when dispersed form their mother plant [81], with their embryonic axis having a higher moisture content than the rest of the seed [82]. As time passes, seed dehydration causes structural damage to the vacuoles, the cytoskeleton and cellular membranes. One of the major causes of recalcitrant seed viability loss has been attributed to permanent damage to cell membranes due to tissue dehydration [81]. Due to the high metabolic activity and high respiration rates of recalcitrant seeds, the generation of free radicals and reactive oxygen species are generated during seed storage, which can damage membranes and generate toxic by-products [83,84], amongst other forms of damage, contributing to further viability loss. The following techniques may extend the viability of recalcitrant palm seeds including coconut; improved handling and transportation, the application of fungicides at the time of collection, and the use of humid, warm storage conditions [28]. Storing recalcitrant seeds imbibed in an oxygen enriched environment (so seeds can maintain their respiratory metabolism) can prolong seed longevity [85]. In addition, seeds can be stored at lower temperatures (but above freezing) to minimize their metabolic activity [81].

For more than 30 years, researchers have been developing protocols to conserve coconut germplasm over the longer-term [10]. If long-term storage is required for coconut germplasm conservation, fruits cannot be used due to their large size [24]. Cryopreservation (storage of meristematic tissues at extremely low temperatures, −196 °C, usually in liquid nitrogen) of their excised embryos is possible and can be considered a better-suited option. Further studies are necessary on fruit storage and longevity to develop ex situ conservation methods.

## 6. Conclusions

Although coconut research has been undertaken for many years around the world, several aspects of the basic germination processes of the coconut fruit remain unknown. Due to such a poor understanding, no well-established and cost-effective protocols have been developed for the germination of coconut varieties in nurseries and in the field. Currently, coconut tissue culture has been developed and may be commercialized soon. Even after commercialization via tissue culture, the traditional methods would still need to support the production of seedlings as the demand for coconut seedlings is increasing every year. If traditional seedling production is to be undertaken in the most cost-efficient manner, then it is crucial to maximize the germination of every fruit batch and promote successful seedling establishment in the field. To achieve this, considerable research is required to characterize fruit morpho-anatomical features to understand the germination process and dormancy mechanisms. If these gaps on germination and dormancy are filled, then cost-efficient production methods can be developed with improved germination rates, uniformity, and total germination. Finally, these understandings will lead to early germination, more uniformity within the population, increased total germination, and better comparison of the growth and development of individual seedlings, to develop a standardized protocol for nursery management practices. Ultimately, this will contribute to increasing the supply of coconut seedlings to meet global demands.

## Figures and Tables

**Figure 1 plants-11-03293-f001:**
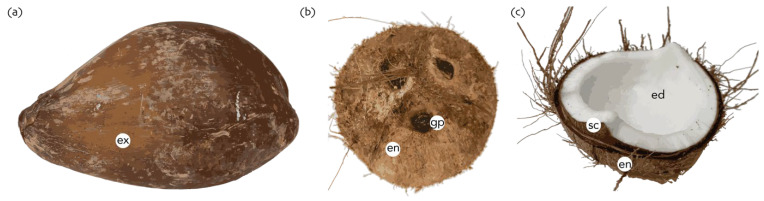
Coconut fruit morphology. (**a**) Whole coconut fruit with the outermost layer visible corresponding to the exocarp (ex); (**b**) a de-husked coconut fruit (exocarp and mesocarp removed), the three coconut eyes (including the germination pore [gp]) can be seen in the endocarp (en) as three dark brown circles; (**c**) coconut cut in half, displaying the inner fruit layer (endocarp [en]), the seed coat (testa [sc]) and endosperm (ed).

**Figure 2 plants-11-03293-f002:**
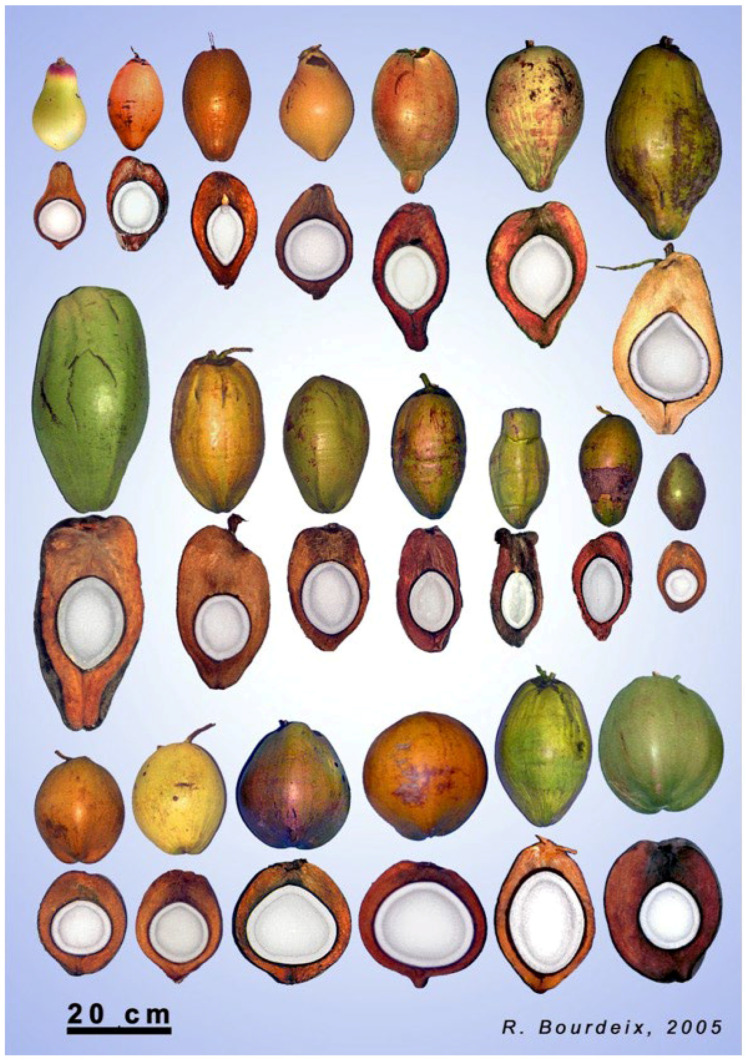
Fruit morphology diversity in different varieties of coconut. Reprinted from Bourdeix Roland, Konan Jean Louis, N’Cho Yavo Pierre. 2005. *Coconut. A guide to traditional and improved varieties*. Montpellier: Diversiflora, 104 p. (Catalys) ISBN 2-9525408-1-0 [34]. Reprinted with permission.

**Figure 3 plants-11-03293-f003:**
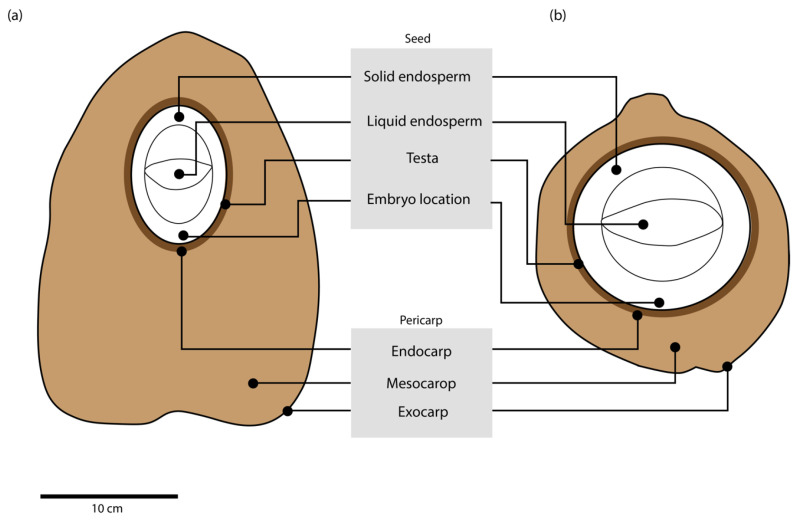
Differences in morphology of (**a**) wild and (**b**) domesticated coconut fruit. Domesticated varieties show a smaller pericarp region, having significantly less mesocarp as compared to wild types. They also have a rounder shape which allows a higher volume of water to be held inside the seed.

**Figure 4 plants-11-03293-f004:**
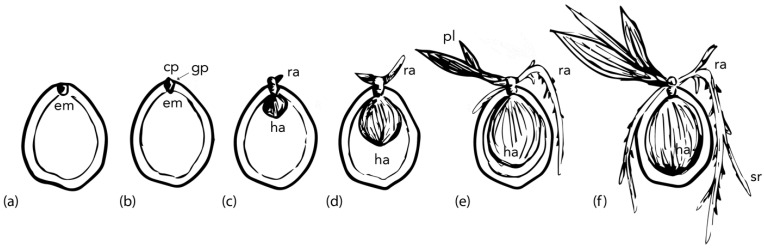
Coconut germination stages: (**a**) Ungerminated coconut fruit. Embryo (em) consists of a short plumular-radicular axis. The plumule consists of a central meristematic zone enclosed by scaly-leaf primordia, which are then surrounded by the cotyledonary petiole. (**b**) In the proximal end of the embryo, the cotyledonary petiole (cp) extends through the germination pore (gp). (**c**) Germinative button formation and radicle emergence (ra) from the cotyledonary petiole. The cotyledonary blade (distal end of the embryo) starts expanding to form a haustorium (ha). (**d**) Radicle elongation, plumule emergence and continuous haustorium growth. (**e**) Secondary roots forming, plumule elongation and first leaves, extensive haustorium growth. (**f**) Secondary root (sr) growth, leaf growth, haustorium completely fills the inner cavity (endosperm).

**Figure 5 plants-11-03293-f005:**
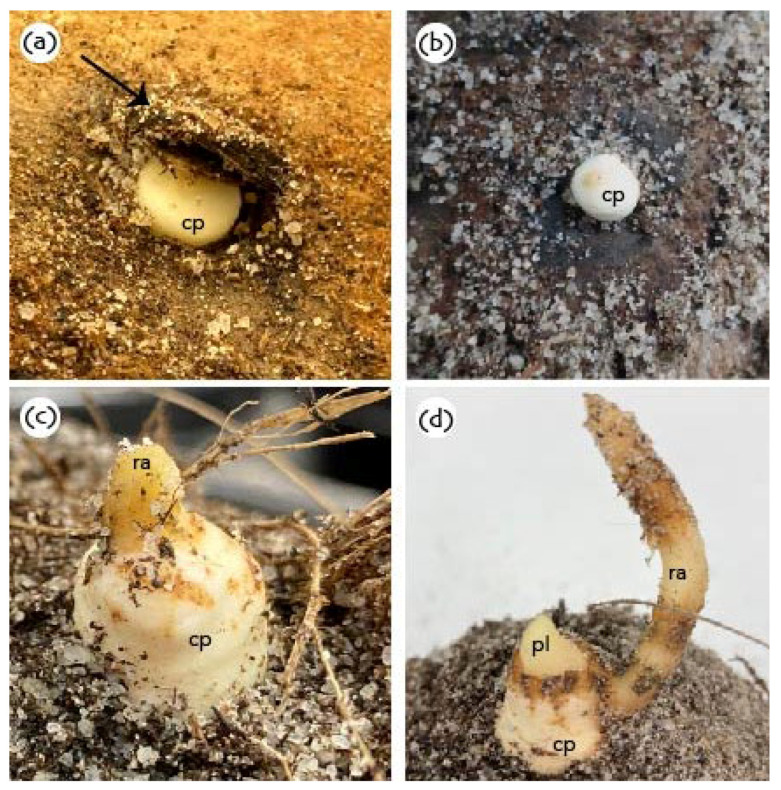
(**a**) Cotyledonary petiole protruding through the endocarp: arrow shows the displacement of the soft eye; (**b**) cotyledonary petiole protruding through the endocarp; (**c**) radicle emerging from the cotyledonary petiole; (**d**) radicle elongation and emergence of the plumule. cp: cotyledonary petiole; ra: radicle; pl: plumule.

**Table 1 plants-11-03293-t001:** Pre-treatment techniques to promote coconut fruit germination.

Type of Pre-Treatment	Benefits	Pre-Treatment	Reference
Mechanical or physical treatment of the fruit	Facilitates moisture penetration into the fruit	De-husking	[27]
		Slicing a portion of the husk	[3,65]
		Chopping husk from both sides	[60]
Physiological treatments by using water or chemicals	Promotes moisture and chemical penetration to stimulate germination	Soaking in water	[3,27,75]
		Soaking in nutrient solution (e.g., KNO_3_ and NaCO_3_)	[27,75]
		Injection of minerals/nutrients (e.g., zinc and copper)	[3]

## Data Availability

Not applicable.

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
