# Peer review of "Fruit Biology of Coconut (Cocos nucifera L.)"

_plants, 2022, doi:10.3390/plants11233293_

Round 1

Reviewer 1 Report

The present review entitled "Fruit biology of coconut (Cocos nucifera L.)" is an interesting review of the state of the art about coconut fruit, however, additional information related with inflorescence development, diversity of fruit, table with studies related with propagation and germination of coconut fruit will enrich this manuscritp. Additional pictures related with coconut fruit diversity.

Author Response

Thank you very much for your reviews on our manuscript. While inflorescence development is an interesting topic, the authors feel it is beyond the scope of this review as we focused on fruit biology of the coconut including germination and dormancy issues. As there is enormous scarcity for good quality planting materials for coconut replanting programs world-wide, this manuscript is mainly reviewed the constraints for the production of planting materials via traditional methods.

Re the suggestion for diversity of fruits and including a table, the diversity of coconut fruit regarding colour, sizes and husk types, was described from lines 127 to 129 and also extended this further from lines 129 to 133. We have also included an excellent image that is showing fruit diversity of coconut (we received permission to use this image from the author). Furthermore, a table has been included (from lines 513 to 517)  that summarizes studies related to propagation and germination of the coconut fruit, as suggested by the reviewer.

Reviewer 2 Report

Looking for the cause/es of low production of coconut is justified and needed in the situation its high demand and in the same time decrease of its production.   Authors of submitted manuscript present  the main processes associated with the morphological features of coconut fruit, its distribution  and germination. Whole paper is based on numerous literature (86 ) andmany studies of scienties from many countries. Many chapters of this paper has informative charcter and many of them contain some suggestion of the  further research which could help to solve the problem with productivity.   

However, 

1)      I did not find any information about fertilization of cocnut cultivation/production. So my question is:  is it in practical cocnut production use practise to apply any nutrients like in the cultivation of many other crops?

2)      Authors menton about coconut soaking in water  so I am interested in the use of other substances to soaking the fruits which have some stimultion effect.

3)      From my own scientific experience effects that some physical methods (like laser irradiation, magnetic field) can increase the germination of plants and vigour of seedlings. May be these methods coud be apply in the case of coconut.

Author Response

Thank you very much for your reviews on our manuscript and for your valuable feedback. Please see below the response for each of the reviewer's points:

1) As this review is focused on the fruit biology of the coconut rather than on coconut production per se, we think that getting into details regarding coconut palm fertilization protocols would be beyond the scope of this manuscript. However, we have included a short description on how fertilization can influence the morphology of the coconut fruit from lines 157 to 160.

2) We explained various methods by which coconut germination can be enhanced, including soaking in water and nutrient solutions from lines 465 to 511. We have also included a table to make this information obvious and useful for the readers (from lines 513 to 517). Other substances are commonly used to enhance germination in seeds, usually by priming seeds with germination enhancing chemicals such as gibberellic acid, but this is usually done in smaller-sized seeds. Further, it would also be beyond the financial capacity of coconut farmers to use such an approach. Unfortunately, authors could not find enough evidence in the literature suggesting the use of other chemicals on coconut fruits.

3) They are very good ideas to use such new approaches to improve the germination. Unfortunately, these kinds of germination promotion are still poorly understood and we have not been able to find information on such technologies used on coconut fruits in the available literature. We are unable to provide suggestions for any new methods in the review paper without any previous evidence on these approaches in coconut fruit. Furthermore, the authors feel such techniques may not be accessible to smallholder coconut farmers as majority of the coconut farmers are smallholders around the world.

Reviewer 3 Report

The content covered by the authors during writing the manuscript is exhaustive, the manuscript may be accepted. Still, grammar and spell check is required. The authors are suggested to incorporate the phytochemistry or chemical constituents involved and possible medicinal values with their mechanism. This will increase the worth of the paper.

Author Response

Thank you very much for your reviews on our manuscript. The authors have read throughout the manuscript for grammar and typo errors. While the chemical constituents of coconut fruits can differ from variety to variety, which is an interesting topic, we did not find any previous studies that the chemical constituent are linked to fruit biology or germination of coconut fruit. Hence, the authors feel it is beyond the scope of this review paper.

Round 2

Reviewer 1 Report

Thanks to the authors for addressing the suggestions for the manuscript.